# Accuracy of the Fluorescence-Aided Identification Technique (FIT) for Detecting Residual Composite Remnants after Trauma Splint Removal—A Laboratory Study

Eva Magni [1,†], Wadim Leontiev [1,†], Sebastian Soliman [2], Christian Dettwiler [1], Christian Klein [3,4], Gabriel Krastl [2], Roland Weiger [1] and Thomas Connert [1,*]

1 Department of Periodontology, Endodontology and Cariology, University Center for Dental Medicine UZB, University of Basel, 4075 Basel, Switzerland
2 Department of Conservative Dentistry and Periodontology, Center of Dental Traumatology, University Hospital of Würzburg, 97080 Würzburg, Germany
3 Department of Restorative Dentistry, Periodontology, Endodontology and Pediatric Dentistry, School of Dental Medicine, Eberhard Karls University, 72074 Tübingen, Germany
4 Private Practice Meller Zahngesundheit, 71332 Waiblingen, Germany
* Correspondence: thomas.connert@unibas.ch
† These authors contributed equally to this work.

**Abstract:** Distinguishing composite remnants from tooth structure after trauma splint removal can be challenging. This study aimed to compare the Fluorescence-aided Identification Technique (FIT) with conventional light illumination (CONV) in terms of accuracy and time required for the detection of composite remnants after trauma splint removal. Ten bovine tooth models containing anterior teeth from 12 to 22 with composite remnants after trauma splint removal were used. These models were examined by 10 students and 10 general dentists. Each examiner assessed the 10 models using CONV or FIT three times with an interval of 2 weeks each using a prototype fluorescence-inducing headlamp with a spectral bandwidth of $(405 \pm 7)$ nm for FIT and a dental unit lamp for CONV. The examiners charted the location of identified composite remnants, and the procedure time needed for each method was recorded. Statistical analysis was performed with R 3.2.2 software with a significance level of $\alpha = 5\%$. FIT was more accurate and less time-consuming than CONV ($p < 0.001$). There were no significant differences between dentists and students concerning accuracy (CONV: $p = 0.26$; FIT: $p = 0.73$). Students performed FIT significantly faster than the dentists ($p < 0.001$). FIT is a quick and reliable method of identifying composite remnants after trauma splint removal.

**Keywords:** composite detection; fluorescence-aided identification technique; titanium trauma splint (TTS) removal

## 1. Introduction

Traumatic dental injuries are common in children and young adolescents [1]. Permanent teeth with root fractures and luxations require careful repositioning or replantation, followed by splinting [2]. The period of splinting varies depending on the type of injury. The splint is removed after one to four weeks in most cases (avulsion, subluxation and lateral, intrusive and extrusive luxation injuries), but may be left in place for as long as 12 weeks after certain injuries, such as root fractures [2]. The Titanium Trauma Splint (TTS, Medartis AG, Basel, Switzerland) was designed for this application. The TTS is a simple appliance that meets contemporary esthetic standards [3,4]. It is bendable and can therefore be adapted to the dental arch. The matt-silver TTS consists of intersecting rhombuses with gaps for the application of composite resin materials (CRM) to bond the splint to the teeth. Tooth-colored composite makes the TTS inconspicuous but is also difficult to differentiate from tooth enamel during splint removal [5].

Modern tooth-colored composite restorations can nearly perfectly imitate the visual appearance of the tooth structure [6]. Fluorescent materials, such as rare earth oxides, are added to the glass fillers of CRM to match the fluorescence properties of the enamel and dentine [7–9]. Nevertheless, the fluorescence emissions of these restorative materials differ from those of natural tooth structure when illuminated at different wavelengths of light [9]. The maximum fluorescence of composite resin materials occurs at a wavelength of $(398 \pm 5)$ nm [9]. The Fluorescence-aided Identification Technique (FIT) takes advantage of the different fluorescence properties of CRM and makes CRM appear brighter than natural tooth structure when illuminated with fluorescent light [10].

Several studies have shown that, compared to conventional illumination, fluorescence illumination improves the identification of composite fillings and selective composite removal on teeth in general and in posterior teeth [10–18]. FIT is a straightforward, fast, non-invasive diagnostic tool with good reliability and operator agreement [10]. Furthermore, FIT facilitates orthodontic bracket debonding [19–21] and the detection of composite restorations in forensics [13,22,23].

Dettwiler et al. showed the use of FIT to be beneficial during the removal of composite bonded trauma splints [5]. Compared to conventional dental lighting, using FIT resulted in significantly fewer composite remnants and less iatrogenic defects [5].

The potential consequences of composite remnants include esthetic impairment due to discoloration of composite margins and plaque retention spots that may result in demineralization of the enamel over time [24,25]. Furthermore, composite remnants may impair the bond strength of later adhesive restorations, which are often required after dental trauma [26–30].

The detection of composite remnants after trauma splint removal by a dentist or dental student who did not remove the splint initially has not been investigated to date. Therefore, the aim of this study was to compare the FIT versus conventional method (CONV) of detecting composite remnants after trauma splint removal in terms of the accuracy and procedure time, as well as with respect to the professional experience and gender of the examiner.

## 2. Materials and Methods

### 2.1. Tooth Model Fabrication and Digitalization

Intact extracted bovine incisors that had been cleaned and stored in 0.5% chloramine-T solution at room temperature directly after extraction until further processing were used to fabricate 10 upper jaw models ($n$ = 10) of anterior bovine teeth 13 to 23 from ProBase polymethyl methacrylate resin (Ivoclar Vivadent AG, Schaan, Liechtenstein). Preoperative surface scans of the teeth were acquired using a CEREC Omnicam and CEREC SW 4.5.1 software (Dentsply Sirona, York, PA, USA). All scans of teeth with unimpaired surfaces were exported to OraCheck 2.13 (Cyfex AG, Zurich, Switzerland) for later superimposition and analysis, according to the study protocol.

### 2.2. Splint Application

A TTS splint (Medartis AG, Basel, Switzerland) was adhesively bonded to teeth 12 to 22 of each bovine model, following a standardized protocol. Briefly, the bonding surfaces were determined using a customized perforated silicon template. Due to the size of the bovine teeth, two bonding sites were placed in the middle third of the crown of each tooth (site diameter: 3 mm). Each bonding site was etched with Ultra-Etch (Ultradent Products Inc., South Jordan, UT, USA) for 30 s; Heliobond (Ivoclar Vivadent AG, Schaan, Lichtenstein) was applied and light-cured for 30 s with a Bluephase 20i light curing unit (Ivoclar Vivadent AG, Schaan, Lichtenstein) at an output intensity of 1200 mW/cm$^2$. Tetric EvoFlow A2 (Ivoclar Vivadent AG, Schaan, Lichtenstein) was applied using the aforementioned silicon template and light cured, as described above. The TTS splint was placed on the applied composite and covered with the same flowable composite while ensuring that the specified size of the bonding site was respected and not exceeded. The tooth models were

stored at room temperature in Ringer solution (B Braun AG, Melsungen, Germany) until further processing.

### 2.3. Splint Removal

The removal of the TTS splints was performed by two dentists with 2 12 tw1elve years of professional experience, respectively. To simulate clinical conditions, the upper jaw model was inserted in the mouth of a dental mannequin (Frasaco GmbH, Tettnang, Germany), which was seated and secured in an Teneo dental chair (Dentsply Sirona, York, PA, USA). Splint removal was performed under standardized lighting conditions using the LEDview dental unit lamp (Sirona, York, PA, USA). The following instruments could be used in splint removal: dental mirrors, a curved dental explorer, a three-way triple air-water syringe, a high-speed contra-angle hand piece (1:5, KaVo Master Series, Biberach, Germany), cylindrical diamond burs (FG 4038, Intensiv SA, Montagnola, Switzerland), and carbide burs (Bonding Resin Remover, H22ALGK 016, Komet Dental, Lemgo, Germany). Postoperative surface scans of the teeth were performed using a CEREC Omnicam with CEREC SW 4.5.1 software (Dentsply Sirona, York, PA, USA). Following splint removal, all scans were exported to OraCheck 2.13 (Cyfex AG, Zurich, Switzerland) for superimposition and further analysis according to the study protocol. The models were stored in Ringer solution (B Braun AG, Melsungen, Germany) at room temperature until further processing.

### 2.4. Composite Remnant Identification

The tooth models were examined for composite remnants by 10 general dentists with 3 to 24 years of professional experience and by 10 dental undergraduate students. The inclusion criteria for participants were normal visual acuity and the absence of color blindness assessed by an Ishihara test [31]. For undergraduate students, the criteria were active participation in the first- or second-year master's degree program in dentistry and an average grade of B or better. Accordingly, exclusion criteria were visual impairment, color blindness, and missing clinical activity. The examiners were instructed to identify and chart all composite remnants using a dental mirror, a curved dental explorer, and a three-way triple air-water syringe by the respective illumination method. All examinations were carried out under standardized lighting conditions in a darkened room illuminated only by artificial light. The tooth surfaces were illuminated with an LEDview dental unit lamp (Sirona, York, PA, USA) for CONV and with a prototype fluorescence-inducing headlamp with a spectral bandwidth of $(405 \pm 7)$ nm (Karl Storz GmbH & Co. KG, Tuttlingen, Germany) for FIT (Figure 1). Each model was assessed in triplicate by each examiner with a break of 14 days in between, so that the participants could not remember previous findings. The investigators additionally recorded the procedure time required for each model. The experiments were performed as follows:

- Examination 1: Evaluation of all 10 models by CONV.
- Examination 2 (14 days later): Evaluation of five models by CONV and five models by FIT.
- Examination 3 (14 days later): Evaluation of all 10 models by FIT.
- All examinations were performed in the same room under the same ambient light conditions (examinations at the same day time with no direct solar irradiation, 500–800 Lux) at the University Center for Dental Medicine Basel, Switzerland.

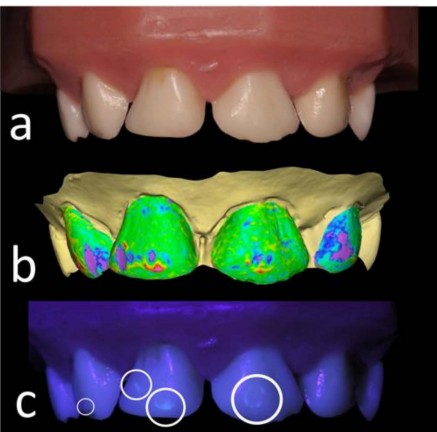

**Figure 1.** Representative model (**a**) illuminated by the conventional illumination (**b**) with visualized surface changes in the OraCheck software (**c**), illuminated by the fluorescence-inducing light source.

### 2.5. Data Analysis

Two experienced dentists (S.S. and T.C.) not involved in the identification of composite remnants evaluated the dental charting results on the basis of volumetric change calculation. Briefly, the pre- and postoperative scans were superimposed using the best-fit method [32]. All splinting sites on the teeth were overlapped independently for more accurate super-imposition. Teeth 13 and 23 were not included in splinting to facilitate the registration of the scans. The OraCheck software color-coded the results for clear visualization of surface changes between scans. Green was used to indicate unchanged sites, blue and violet for substance loss, and yellow, red, and pink for excess material (Figure 1). The visualized areas were then analyzed with the OraCheck linear and volumetric measurement tool. The results were used to make a transparent solution template for each model that could be superimposed over the dental charts for grading purposes.

Each dentist evaluated the correctness of identification of composite remnants (COI) for each tooth on a scale of 1 to 7 (Table 1), where the score 1 represented "correct" identification and the scores 2 to 7 were classified as "false" identification for dichotomous analysis. If the two dentists disagreed on a site, it was jointly re-evaluated by both dentists until a consensus was reached. The 10 models with four teeth each were assessed for composite remnants by 20 examiners (10 dentists and 10 students), yielding a total of 800 assessments.

**Table 1.** Correctness of identification of the composite remnants (COI) scoring system.

| | |
|---|---|
| 1: FD | Fully correct detection |
| 2: PD | Partly detected |
| 3: ND | Not detected |
| 4: FP | False positive entirely (no composite in situ) |
| 5: FD+ | FD + false positive |
| 6: PD+ | PD + false positive |
| 7: ND+ | ND + false positive |

### 2.6. Statistical Analysis

As descriptive measures for categorical (years of professional experience) and binary parameters (CONV vs. FIT; dentist vs. student; male vs. female examiner), count and relative frequencies were used. For continuous measures (recorded time for examination), mean values, standard deviation, median, minimum, and maximum were calculated (Table 2) and normality was checked with quantile-quantile-plots (normality-plots, q-q-plots).

**Table 2.** Distribution of correctness of identification (COI) grades 1–7 and their dichotomous categorization as "correct" (score 1) and "false" (score 2–7) for examination 1 (CONV), 2 (CONV and FIT), and 3 (FIT).

|  | COI Score | Examination | | |
|---|---|---|---|---|
|  |  | **1** | **2** | **3** |
| CONV | 1 | 209 | 95 | / |
|  | 2 | 93 | 87 | / |
|  | 3 | 147 | 92 | / |
|  | 4 | 105 | 31 | / |
|  | 5 | 58 | 34 | / |
|  | 6 | 45 | 18 | / |
|  | 7 | 143 | 43 | / |
|  | Correct | 209 | 95 | / |
|  | False | 591 | 305 | / |
| FIT | 1 | / | 354 | 714 |
|  | 2 | / | 5 | 27 |
|  | 3 | / | 15 | 22 |
|  | 4 | / | 10 | 12 |
|  | 5 | / | 13 | 20 |
|  | 6 | / | 0 | 1 |
|  | 7 | / | 3 | 4 |
|  | Correct | / | 354 | 714 |
|  | False | / | 46 | 86 |

For inter-rater-reliability and retest-reliability, Cohens Kappa with 95%-confidence interval (CI) was calculated.

Comparison of the binary outcome ("correct" (rating 1) vs. "false" (rating 2–7)) between examination 1 and examination 3 was calculated with the McNemar test and with Fisher's Exact test for examination 2, respectively.

For comparison of the binary outcome "correct" (grading 1) vs. "false" (grading 2–7) between students vs. dentist, male vs. female examiners, and examiners with a variable professional experience, the Fisher's Exact test was utilized, excluding data from examination 2 due to a small sample size.

The recorded time was compared for CONV and FIT using the nonparametric Wilcoxon test, comparing examination 1 and examination 3 and the measured values within examination 2. Likewise, the time needed for examination was compared for dentists vs. students for CONV (examination 1) and FIT (examination 3), using the Wilcoxon test.

The level of significance was set at $\alpha = 5\%$ in all analyses, which were performed utilizing the software R version 3.2.2.

## 3. Results

### 3.1. Correctness of Identification of Composite Remnants

Overall differences in the correctness of identification of composite remnants between CONV vs. FIT were significant ($p < 0.001$) in examinations 1 and 3, as well as in examination 2 ($p < 0.001$). The distributions of COI grades 1–7 and their dichotomous categorization as correct vs. false identification in examinations 1, 2, and 3 are presented in Table 2. There were no significant differences in COI between dentists vs. students ($p = 0.26$ for CONV; $p = 0.73$ for FIT), male vs. female examiners ($p = 0.42$ for CONV; $p = 0.25$ for FIT), or years of examiner experience ($p = 0.74$ for CONV and $p = 0.46$ for FIT). The relative frequencies of correct identification of composite remnants in the various groups are presented in Figure 2.

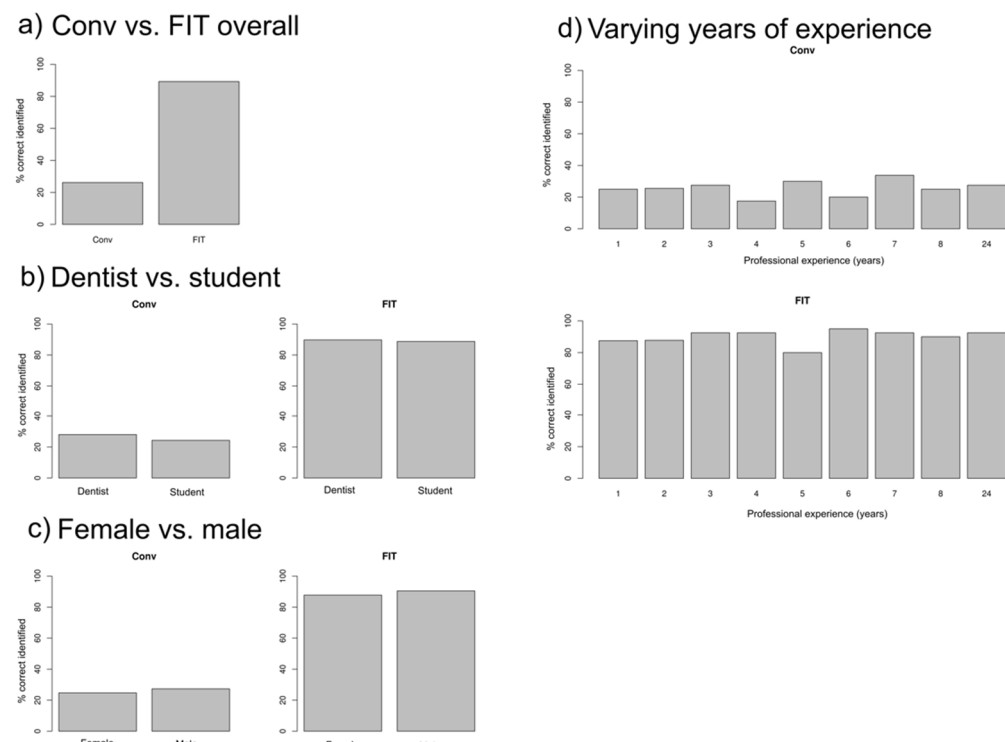

**Figure 2.** Relative frequencies of correctness of identification (COI) grade 1 ("correct") for (**a**) CONV vs. FIT overall, (**b**) dentists vs. students for CONV and FIT, (**c**) female vs. male examiners for CONV and FIT, and (**d**) years of dental experience for FIT and CONV.

*3.2. Procedure Time*

Differences in the procedure times for CONV vs. FIT were significant ($p < 0.001$ for examination 1 vs. 3; $p < 0.001$ for examination 2). There were no significant differences in procedure time between dentist and students for CONV ($p = 0.17$). However, the procedure time for the identification of composite remnants by FIT was significantly shorter for students compared to dentists (22 s vs. 25 s per tooth; $p < 0.001$). The measured procedure times are detailed in Table 3.

**Table 3.** Procedure time required for the identification of composite remnants by CONV vs. FIT.

| Examination | | n | Mean $\pm$ Standard Deviation [s] | Median [s] | Min [s] | Max [s] |
|---|---|---|---|---|---|---|
| 1 | CONV | 800 | 117 $\pm$ 9 | 120 | 70 | 120 |
| | FIT | | No assessment | | | |
| 2 | CONV | 400 | 118 $\pm$ 5 | 120 | 90 | 120 |
| | FIT | 400 | 23 $\pm$ 5 | 22 | 10 | 38 |
| 3 | CONV | | No assessment | | | |
| | FIT | 800 | 24 $\pm$ 5 | 24 | 10 | 38 |

*3.3. Inter-Rater-Reliability*

Inter-rater-reliability was high ($\kappa = 0.98$, 95%—CI 0.98–0.99). For retest-reliability, the agreement was fair for CONV ($\kappa = 0.34$, 95%—CI 0.28–0.40) and moderate for FIT ($\kappa = 0.43$, 95%—CI 0.30–0.56).

**4. Discussion**

The main objective of this study was to compare the accuracy and procedure time of the Fluorescence-aided Identification Technique (FIT) and the conventional illumination

method (CONV) of detecting composite remnants after trauma splint removal when performed by dentists and students. CEREC surface scanning of the teeth before and after splint removal was used as the gold standard for remnant identification. Additional data on the impact of examiner experience and gender on the accuracy and procedure time, as well as the inter-rater-reliability and retest-reliability of the two methods were also collected. The detection of composite remnants after trauma splint removal by a dentist other than the operator who removed the splint initially is a clinically relevant situation that has not been investigated yet.

The detection of composite remnants after trauma splint removal was more accurate and faster with FIT compared to conventional illumination. This is in line with the results of previous studies where FIT was superior to the conventional method of composite remnant identification with regard to accuracy and procedure time [10,13,15–18]. Contrary to expectations, the dental students in this study performed the FIT method faster than the dentists. Although the difference in procedure time was statistically significant, it most probably is not clinically relevant. Furthermore, the inter-rater-reliability results showed good consensus between the examiners. Conversely, the retest reliability was low. Moreover, examiner experience did not have an impact on the results. The gender of the examiner is known to influence color shade matching quality [33] but did not have an impact on the results of the study.

A bovine incisor model was used in the present study due to difficulties in acquiring enough suitable human front teeth. Most extracted human anterior teeth are inappropriate for this type of study because of the presence of coronal restorations or carious defects. The use of bovine teeth enabled a standardized study protocol. However, the use of non-human teeth differs significantly from the clinical situation, which may have influenced the results. All teeth were extracted on the same day and cleaned and stored in the same manner.

The template used to evaluate the results of composite remnants charting was generated by superimposing the intraoral scans with the OraCheck software. Intraoral scanners have a deviation tolerance of 20 μm for single teeth, 35 μm for quadrants, and 50 to 80 μm for full-arch scans [34–36]. To achieve precise superimposition, we used teeth 13 and 23 as unchanging reference points and kept the accuracy setting for superimpositions very high in light of the very small changes observed in this study (level "0.00 mm < distance ≤ 0.1 mm" over 95%) [5].

The fluorescence of the resin composite material used in this study was noticeably greater than that of natural tooth substance. However, some resin composites have even stronger fluorescence [9]. The choice of composite might have influenced the results of this study. While different composite resin products fluoresce at different intensities, the FIT method is suitable for identification of most of the commercially available composites [9,37]. The clinician should take the fluorescence properties of the composite used for trauma splint fixation into account in regard to the application of the FIT method.

Several studies have shown that different composite resins are subject to aging processes and a related decrease in fluorescence properties [38–41]. Any effects of aging in the present study were most likely negligible due to the short storage time in particular and the short period of time that splints remain in the mouth in general. The composite identification experiments in the current study were performed shortly after the precursor study by Dettwiler et al. [5].

FIT was performed using a prototype headlamp. A recent study showed that various light sources with a wavelength of approximately 400 nm are suitable fluorescence-inducing devices, which are superior to the conventional method [18]. FIT can be performed by affordable and easy to handle devices, such as headlamps that illuminate the whole oral cavity.

The present study was designed to simulate clinical conditions as closely as possible to increase the reliability of the results. Moreover, a template to mark the sites of application of adhesive composite resin in order to standardize the trauma splint bonding procedure was used. This might have influenced the results of the study as bonding composites are more

randomly placed on the teeth in routine clinical practice. All examinations were performed under standardized lighting conditions, in a darkened room to obtain reliable data.

Accurate differentiation of dental composite materials from natural tooth substance enables the minimally invasive and nearly complete removal of composite remnants. This is clinically relevant since composite remnants may lead to impaired esthetics and plaque accumulation. Furthermore, no drying of the teeth is necessary with the FIT method, whereas frequent drying of the teeth is needed during the removal of composite remnants by the conventional method. This has important clinical implications since constant drilling and drying significantly increases the procedure time. Moreover, when using high-speed rotary instruments, continuous water cooling is crucial to maintaining the integrity of the pulpal tissues, particularly in traumatized teeth [42–44].

## 5. Conclusions

FIT is a fast and reliable method for detecting composite remnants after trauma splint removal. Moreover, the accuracy and procedure time of FIT is not dependent on the level of professional experience of the dentist.

**Author Contributions:** Conceptualization, T.C., G.K. and C.D.; methodology, T.C. and R.W.; formal analysis, R.W.; investigation, C.D. and S.S.; resources, C.K.; data curation, E.M. and W.L.; writing—original draft preparation, E.M. and W.L.; writing—review and editing, E.M., W.L., S.S., C.D., C.K., G.K., R.W. and T.C.; visualization, W.L. and C.D.; supervision, T.C.; project administration, T.C.; funding acquisition, T.C. and C.D. All authors have read and agreed to the published version of the manuscript.

**Funding:** This research was funded by the Swiss Dental Association (SSO), grant number 292-16.

**Institutional Review Board Statement:** This investigation was conducted in conformity with the principles set for the WMA's Statement on Animal Use in Biomedical Research.

**Informed Consent Statement:** Not applicable.

**Acknowledgments:** The authors would like to thank Daniela Keller for the help with the statistical analysis. The authors also like to thank the Swiss Dental Association (SSO) for funding this work through SSO Research Grant 292-16.

**Conflicts of Interest:** The authors declare no conflict of interest.

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
