# Peer review of "Accuracy of the Fluorescence-Aided Identification Technique (FIT) for Detecting Residual Composite Remnants after Trauma Splint Removal—A Laboratory Study"

_applsci, doi:10.3390/app121910054_

Round 1

Reviewer 1 Report

I believe that the article has a topic of interest, with applicability in dental practice, especially for orthodontists who do not always manage to completely remove the composite resin, these remnants becoming ecological niches for bacterial plaque and an aesthetic problem.

The way the material was structured and presented is clear and concise.

In the introduction and discussions, I found elements and references that highlight the importance of the topic as well as the results in relation to the specialized literature.

In the material and methods, the working method is described in detail and the results are presented both descriptively and graphically in a clear manner.

Author Response

Dear Editor, dear Reviewers,

We would like to express our gratitude to you for the swiftness in responding and the cogent and constructive comments. We are delighted to inform you that it has been possible to address each point raised in your feedback in a way we deem adequate. Please do not hesitate to contact me if you have any further queries. For your time and consideration, many thanks.

Response to Reviewer 1:

Thank you very much for your time and your comments. We are very pleased about your positive feedback.

Reviewer 2 Report

Thank you for allowing me to review this article. The purpose of the work was to compare the FIT versus conventional method (CONV)

of detecting composite remnants after trauma splint removal in terms of accuracy and procedure time, as well as concerning the professional experience and gender of the examiner.

I have a few suggestions for authors:

1. What type of study is this, when was it conducted, where was it conducted, and who approved it?

2. Why is paragraph 2.3 Splint removal put in italics?

3. The authors stated, "...by ten dental undergraduate students with one or two years of experience." - what experience, what year of study, and what kind of students are they according to grades?

4. What are the inclusion and exclusion criteria for respondents?

5. Are there study limiting factors? For example, what is the strength of the study?

6. Differences in years of experience, gender, and method should be made separately for both groups or look at the same by correlation?

Author Response

Dear Editor, dear Reviewers,

We would like to express our gratitude to you for the swiftness in responding and the cogent and constructive comments. We are delighted to inform you that it has been possible to address each point raised in your feedback in a way we deem adequate.

Please do not hesitate to contact me if you have any further queries. For your time and consideration, many thanks.

Please find our detailed point-by-point response below. Additionally, major changes are highlighted in the main document.

Response to Reviewer 2

Comments and Suggestions for Authors

Thank you for allowing me to review this article. The purpose of the work was to compare the FIT versus conventional method (CONV) of detecting composite remnants after trauma splint removal in terms of accuracy and procedure time, as well as concerning the professional experience and gender of the examiner.

I have a few suggestions for authors:

Thank you for revising our manuscript and for your valuable suggestions.

  1. What type of study is this, when was it conducted, where was it conducted, and who approved it?

It is a laboratory study (P1, L4).

We added following paragraph:

All examinations were performed in the same room under the same ambient light conditions (examinations at the same daytime with no direct solar irradiation, 500–800 Lux) at the University Center for Dental Medicine Basel, Switzerland. (P3, L144)

The use of bovine teeth was conducted in conformity with the principles set for the WMA’s Statement on Animal Use in Biomedical Research. (P8, L306)

  1. Why is paragraph 2.3 Splint removal put in italics?

Thank you, this has been corrected accordingly.

  1. The authors stated, "...by ten dental undergraduate students with one or two years of experience." - what experience, what year of study, and what kind of students are they according to grades?

&

  1. What are the inclusion and exclusion criteria for respondents?

Thank you very much for this comment. We added the following:

The inclusion criteria for participants were normal visual acuity and the absence of color blindness assessed by an Ishihara test (31). For undergraduate students, the criteria were active participation in the first- or second-year master’s degree program in dentistry and an average grade of B or better. Accordingly, exclusion criteria were visual impairment, color blindness, and missing clinical activity.

  1. Are there study limiting factors? For example, what is the strength of the study?

Thank you for this note. The choice of the composite resin material and the fact that we conducted a laboratory study might have influenced the results, these factors are described in the discussion (P7, L262; P7, 281). Also, we mentioned in our manuscript that we used non-human teeth for our study. (P7, L249).

  1. Differences in years of experience, gender, and method should be made separately for both groups or look at the same by correlation?

Thank you for this valuable comment, which we also considered. However, we decided against it, as there were no further significant findings in this regard. In our opinion, it would then also have become very cluttered for the reader. The statistics were carried out by a professional and recognized statistician.

We understand your argument very well, but we hope that you can agree with the above explanation.

Again, we would like to thank the referees and the editor for their valuable comments, which further improved the clarity and transparency of the study report. We hope that you are satisfied with the revised manuscript.

Reviewer 3 Report

Thanks for the smart idea and the perfect standardized execution

Author Response

Dear Editor, dear Reviewers,

We would like to express our gratitude to you for the swiftness in responding and the cogent and constructive comments. We are delighted to inform you that it has been possible to address each point raised in your feedback in a way we deem adequate. Please do not hesitate to contact me if you have any further queries. For your time and consideration, many thanks.

Response to Reviewer 3:

Thank you very much for your time and your comments. We are very pleased about your positive feedback.

Reviewer 4 Report

Very well done. The only question I have is about the methodology. 

Why was the FIT done 14 days later? Would you explain?

Author Response

Dear Editor, dear Reviewers,

We would like to express our gratitude to you for the swiftness in responding and the cogent and constructive comments. We are delighted to inform you that it has been possible to address each point raised in your feedback in a way we deem adequate. Please do not hesitate to contact me if you have any further queries. For your time and consideration, many thanks.

Response to Reviewer 4:

Thank you for your time and the positive feedback.

We chose to repeat the experiment after 2 weeks so that the participants could not fill in the dental charts from memory. Repeating the experiment immediately probably would have distorted the results of the fit group.

Revised text:

Each model was assessed in triplicate by each examiner with a break of 14 days in between, so that the participants cannot remember previous findings. (P3, L136)